# Dynamics of Active Brownian Particles in Plasma

**DOI:** 10.3390/molecules26030561

**Published:** 2021-01-21

**Authors:** Kyaw Arkar, Mikhail M. Vasiliev, Oleg F. Petrov, Evgenii A. Kononov, Fedor M. Trukhachev

**Affiliations:** 1Joint Institute for High Temperatures, Russian Academy of Sciences, 125412 Moscow, Russia; akarkyaw53@gmail.com (K.A.); ofpetrov@ihed.ras.ru (O.F.P.); gadvin@yandex.ru (E.A.K.); ftru@mail.ru (F.M.T.); 2Moscow Institute of Physics and Technology, 141701 Dolgoprudny, Russia

**Keywords:** active particles, Brownian motions, Janus particle

## Abstract

Experimental data on the active Brownian motion of single particles in the RF (radio-frequency) discharge plasma under the influence of thermophoretic force, induced by laser radiation, depending on the material and type of surface of the particle, are presented. Unlike passive Brownian particles, active Brownian particles, also known as micro-swimmers, move directionally. It was shown that different dust particles in gas discharge plasma can convert the energy of a surrounding medium (laser radiation) into the kinetic energy of motion. The movement of the active particle is a superposition of chaotic motion and self-propulsion.

## 1. Introduction

In work [1], the following definition of an active particle is given: “Differently from passive Brownian particles, active particles, also known as self-propelled Brownian particles or micro-swimmers and nano-swimmers, are capable of taking up energy from their environment and converting it into directed motion”. Recently, the attention of the research community studying particles suspended in liquids has been attracted by the so-called “active” particles. Such particles exhibit a modification of Brownian motion due to the effects of autonomous self-propulsion (for example, in the case of bacteria) or due to the special nature of their shape or surface, which makes their properties anisotropic. In the simplest cases, the reason for this behavior is the presence of thermal or chemical gradients on the particle surface. Particles in plasma can become active due to the action of phoretic forces (for example, photophoretic force from laser radiation [2,3]), the “rocket” effect [4], the “negative” ion drag force [5] and others. A very interesting class of artificial swimmers is Janus particles [6,7].

Beyond fundamental questions of physics, chemistry and biology, active matter also presents numerous applied opportunities, from the efficient delivery of drugs to the design of “living” materials with structures and functionalities that cannot be attained in passive materials [8,9].

In the presented experiments, we studied the motions of dust spherical micro-particles in the plasma of RF discharge, generated at a frequency of 13.56 MHz and discharge power W_RF_ = 11.8 W at pressure *p* = 3.5 Pa. We used three typesof particle: melamine-formaldehyde (MF) particles, MF particles entirely coated with copper, and MF particles partially coated with iron, also called “Janus particles”. It is shown that the nature of the motion strongly depends on the type of particle used. The novelty of this work lies in comparing the motion of particles with different surface properties, which are placed in the same plasma medium.

## 2. Data Analysis and Discussion

### 2.1. Materials

In the experiments, we used three types of monodisperse MF spheres with a diameter of 10 μm: the first type was without any coating (Figure 1a); the second was with a surface coated by copper (Figure 1b); and the third one was a “Janus” particle, with its surface partiallycoated with iron (Figure 1c,d). The method for manufacturing Janus particles is described below in Section 3.1. According to the known models [10,11], the charge of the particles can be assumed to be the same. Indeed, the magnitude of the charge of a particle immersed in gas discharges depends on the surface area. For isolated particles, the simple orbital-motion–limited (OML) approximation [10] can be used.

### 2.2. Analysis of Particle Trajectories and Discussion

In order to get a basic understanding of the differences between the motion of regular particles and Janus particles, a good approach is to compare the trajectories of single particles of approximately equal radius in the same environment. In this way, the activity mechanism that would lead the system out of equilibrium [6] can be realized in RF plasma with a ring trap under the influence of laser radiation through the two following phenomena:Photophoresis due to a temperature gradient on the particle surface, which is an asymmetric neutral drag force caused by a temperature difference (for all particles);Photophoresis due to different accommodation coefficients, which is a neutral drag force caused by different accommodation coefficients of MF and iron (for Janus particles).

Photophoresis is a force which is related to light radiation. Temperature gradients, along with the rotation of the particle about the axis, can create forces that induce its active motion [12]. For a Janus particle illuminated by strong laser light, temperature gradients on the particle’s surface can also cause active motion due to the selective heating of the various parts of its metallic cover. This leads to a strong thermophoretic motion that can be controlled by the changing of the laser radiation power, as has been shown in the case of Au-capped colloidal particles [1]. Thus, photophoresis is an activity factor both for particles with a homogeneous surface and for Janus particles.

We observed single particle movements under exposure of the argon laser with different values of radiation power. The particles were injected into a discharge chamber, where they started to levitate in the ring trap near the lower electrode (see [13]). The video recorded in the experiment was processed by a special program which determined the positions of the dust particle on video frames with sub-pixel accuracy. As a result of the processing of video recordings, we obtained trajectories for different types of particles, shown in Figure 2. The corresponding video is presented in the Appendix A.

All particles tend towards the center of the symmetrical electrostatic trap at low laser power. This state corresponds withthe minimum potential energy of the particles. The nature of the movement of different types of particles has significant differences. In the case of symmetric particles, the motion is circular with some stochastic (Brownian) components. The mechanism of this motion can be associated with the spin-orbital resonance [12]. Unlike uncoated MF particles, copper-coated particles effectively absorb laser radiation [14]. This, in turn, leads to a noticeable growth in their kinetic energy, so that the radius of particle’s rotation in the circular electrostatic trap also increases. The movement of Janus particles is much more complicated. The shape of their trajectory strongly depends on the power of laser radiation. There are practically no circular tracks, although some sections of the trajectories were in the form of arcs at high laser powers. A kink was often observed between adjacent arcs.

The observed characteristics of motion at a high intensity of laser radiation are in good agreement with the theoretical model [12].The motion of homogeneous particles corresponds to “noise-free motion mode of a circle swimmer with constant self-propulsion in a constant spatial trap” (see Figure 2a,b from [12]), while the motion of Janus particles partially corresponds to “noise-free motion mode of a circle swimmer with temporally varying self-propulsion force in a constant spatial trap”(Figure 3 from [12]). Nevertheless, real trajectories of Janus particles differ from the ideal ones described by the model [12]. In both cases, the trajectories consist of arcs; however, the connection between arcs in the experiment is more chaotic than in the model [12].

To analyze the motion of particles, we calculated their mean square displacement (MSD), which is described by the formula $\langle r^{2}\left(t\right)\rangle =\langle {\left[r\left(t\right)-r\left(0\right)\right]}^{2}\rangle$, where vector ***r***(0) is the initial position of the particle, and vector ***r***(*t*) is the position of the particle at time *t* [15]. In the single-particle case, averaging is carried out over time. The <*r*(*t*)>^2^ plot is shown in Figure 3. It was observed experimentally that the effect of laser radiation with a power of 1600 mW on an uncoated particle was the smallest in comparison with other types of particle. At the same time, for a MF particle with a copper coating and for a Janus particle, the effect of laser radiation of the same power led to an increase in the kinetic temperature and enlarging the region of their motion.

One can see in Figure 3 that in the short time scale, the <*r*(*t*)>^2^ plot demonstrates a ballistic regime with an asymptotic of ~*t*^2^ for all types of particles. In the mesoscale, at time ~1 s, the particle reaches the edge of the potential well (at the corresponding laser radiation power), so that we observe a “confined in the trap” mode of motion (see [16], Figure 4b at a = 1), which is shown in the graph as a “plateau”. There is no asymptotic of ~*t*^3/2^ typical for active Brownian motion, in contrast to classical experiments for extended structures [1].

In order to characterize the rotational motion of various types of particle, we also measured their average linear displacement versus time (see Figure 4). For non-absorbing uncoated MF particles, the rotation frequency in the electrostatic trap is practically independent of the laser radiation power. In this way, even at a maximum laser power of 1600 mW, the amplitude and kinetic energy of motion (see Figure 5) approximately correspond to the characteristic motion of particles with a coating (full coverage and Janus) at a minimum laser power.

In contrast to uncoated particles, particles with a copper shell demonstrate a significant dependence on the magnitude of the rotation amplitude vs. various powers of the acting laser radiation. Moreover, a change in the laser power for ~35% (from 1000 to 1600 mW) leads to a change in the rotation frequency by ~5%.

For Janus particles, a change in the amplitude is observed not only with a change in the power of laser radiation, but also at its fixed values, which is associated with an essential anisotropy of their properties. This happens due to a change in the motion direction, for example, as a result of spin rotation or“tumbles” made by a particle, etc. (see typical trajectories in Figure 2).

The kinetic energy of uncoated MF particles is low in the entire range of laser radiation power, since they weakly absorb light (see Figure 5). For MF particles with a continuous copper shell, the increase in kinetic energy can be explained by the growth of the photophoretic force due to the heating of the absorbing surface of the particle. For Janus particles, an increase in the kinetic energy of particle motion is also observed with the growth of the power of laser radiation; however, absolute values are lower compared with Cu-covered particles. This, most likely, can be associated with the weaker absorption of radiation by its “island” coating.

## 3. Experimental Setup

A scheme of the experimental setup for the studies of active particle motion in a RF discharge is presented in Figure 6. The experiments were carried out in a gas-discharge chamber, which had optical windows for laser radiation input and observations. The chamber was pumped out, after which it was filled with argon to a pressure of 3.5 Pa. From the RF generator through an impedance-matching device, a voltage of 300 V with a frequency of 13.56 MHz was applied to flat, horizontally oriented electrodes, resulting in a discharge being ignited. To form an electrostatic trap, a copper ring 35 mm in diameter was placed on the lower electrode [13]. The diameter of the ring was selected so that dust particles were reliably held by the trap, but at the same time could perform a Brownian motion of large amplitude. Dust particles were injected into the discharge chamber through a hole in the upper electrode. In the same way as it was done in our recent study [17], in order to visualize and heat the particles, we illuminated particles by a homogeneous beam of an argon laser with a wavelength of 514 nm. The position of the dust particles was recorded by a high-speed video camera, with a frame rate of 400 fps and a resolution of 1440 × 1440 (14.6 μm/pixel), located above the electrode.

### 3.1. Janus Particle Synthesis

Janus particles were produced in the experimental setup in the following way: original MF particles 10 μm in diameter were placed on a special substrate, so that they covered the surface uniformly, after which the substrate was installed on the lower electrode. The gas discharge chamber was evacuated and filled with a plasma-forming gas argon to a pressure of 5 Pa. The plasma of a capacitive high-frequency discharge was generated with an applied power of W_load_ = 15 W, while the reflected power was W_ref_ = 3.2 W. The particles were exposed in the plasma of the high-frequency discharge for 315 min, so that their surface became modified.

After this we carried out a scanning electron microscopy (SEM) analysis of the surface of the original MF particles and particles from the substrate after exposure in plasma. The SEM method allowed us to obtain an image of the surface of the material under study with a high spatial resolution (0.4 nm), as well as to carry out X-ray spectral microanalysis to obtain the elemental composition of the material under study. The resulting images of the original and modified particles are shown in Figure 1. According to the obtained data, the original uncovered MF particles had an undeveloped surface structure characteristic for polymers, while the modified particles exhibited partial surface destruction with the presence of iron in its composition. Thus, under the action of low-energy (*E_i_*~100 eV) plasma-forming gas ion flows, erosion of a part of the particle surface that was in contact with the plasma occurred. At the same time, there was a modification of the particle surface due to the deposition on it of the products of erosion of steel electrodes and the walls of the gas-discharge chamber. As a result of the modification of MF particles in the RF discharge plasma, Janus particles with anisotropy of properties were obtained. The modified surface area comprised ≤50% of the entire particle surface.

## 4. Conclusions

The active Brownian motion of single dust particles induced by laser radiation in an electrostatic symmetric trap of an RF discharge is studied experimentally. It is shown that dust particles in gas discharge plasma can convert the energy of the surrounding medium (laser radiation) into the kinetic energy of motion. A very important step of this work is the comparison between MF, MF copper-coated, and Janus particles in plasmas. Absorption of laser radiation by the metal surface of the particle creates a thermophoretic force, which in turn makes the particle active. We observed experimentally the active Brownian motion (directed or irregular) caused by the action of thermophoretic force for various laser intensities. Various patterns of particle motion are detected.

## Figures and Tables

**Figure 1 molecules-26-00561-f001:**
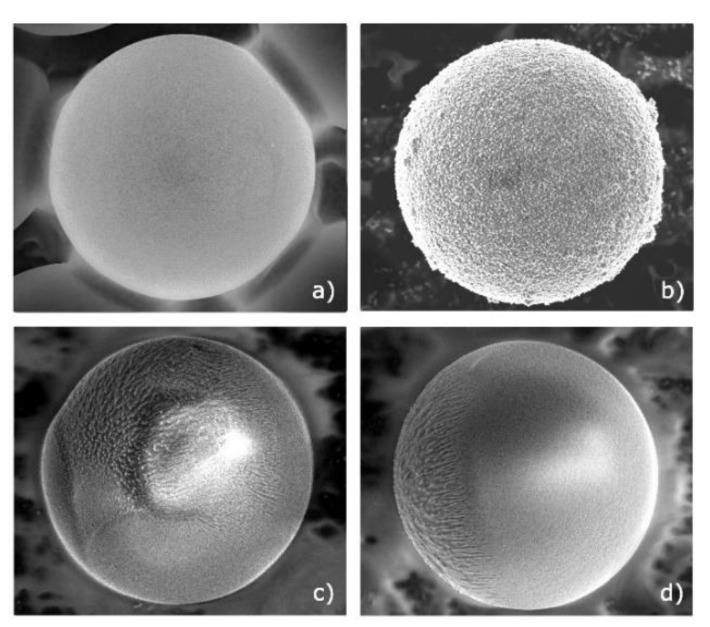
SEM photographs of a spherical monodisperse melamine-formaldehyde (MF) particle with a diameter of 10 μm: (**a**) without coating; (**b**) with a surface coated by copper; (**c**,**d**) Janus particles with iron partly-coated surfaces.

**Figure 2 molecules-26-00561-f002:**
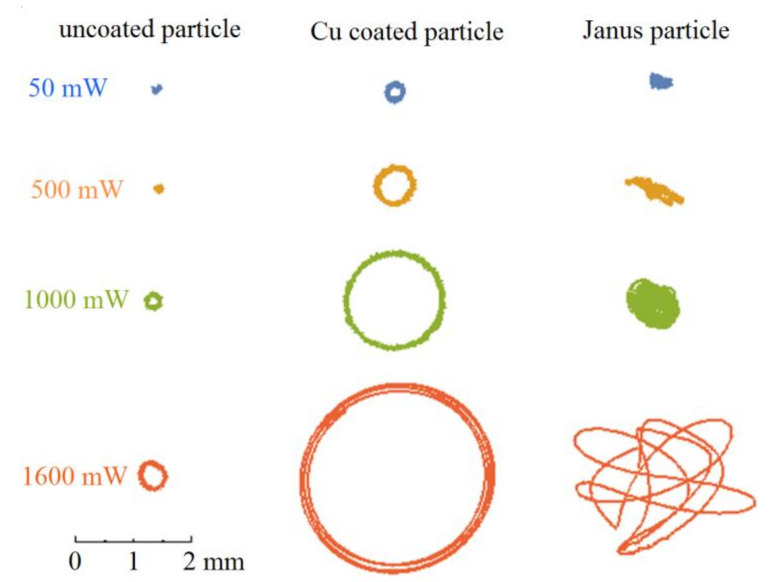
Trajectories of a particle during 5 s, levitating in the near-electrode region of a RF discharge under action of laser radiation with a power of 50, 500, 1000 and 1600 mW. Left row—uncoated particle; center row—Cu-coated particle; right row—Janus particle.

**Figure 3 molecules-26-00561-f003:**
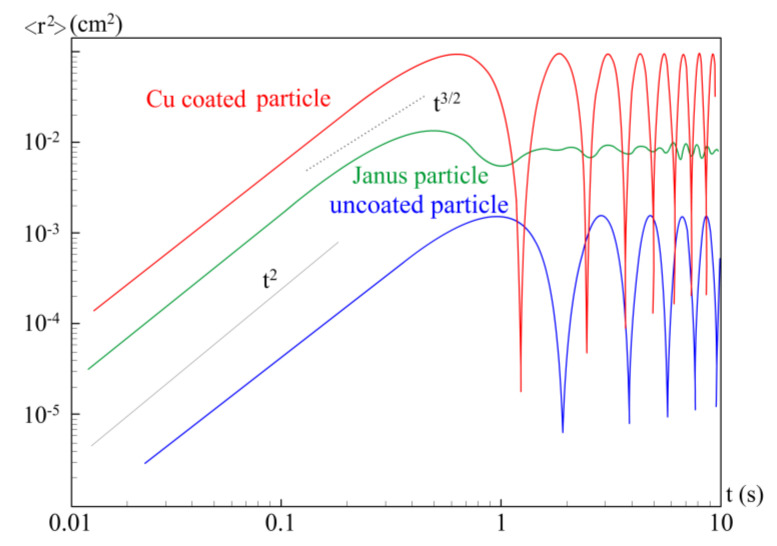
Mean square displacement of motion under exposure of laser with radiation power 1600 W for uncoated MF particles (blue), for copper coated MF particles (red) and Janus particles (green).

**Figure 4 molecules-26-00561-f004:**
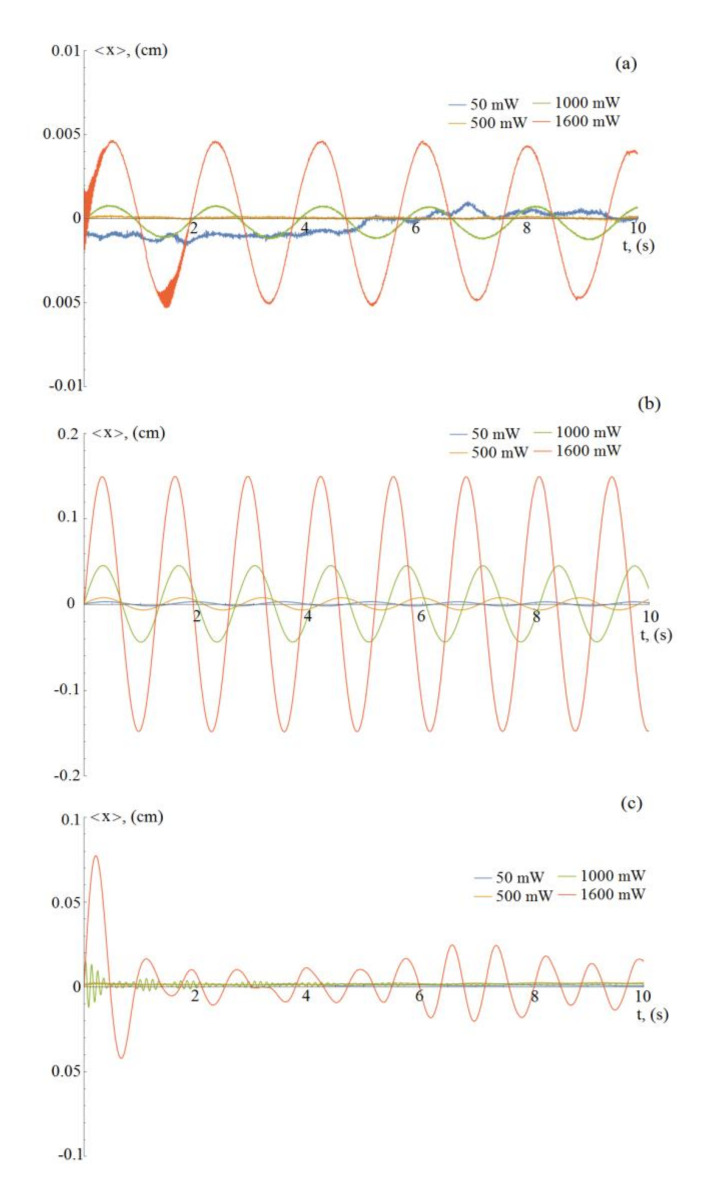
Linear displacements of a single dust particle during 10 s at a gas (argon) pressure of 3.5 Pa, under influence of laser radiation with a power of 50, 500, 1000 and 1600 mW: (**a**) uncoated particle, (**b**) Cu-coated particle, (**c**) Janus particle.

**Figure 5 molecules-26-00561-f005:**
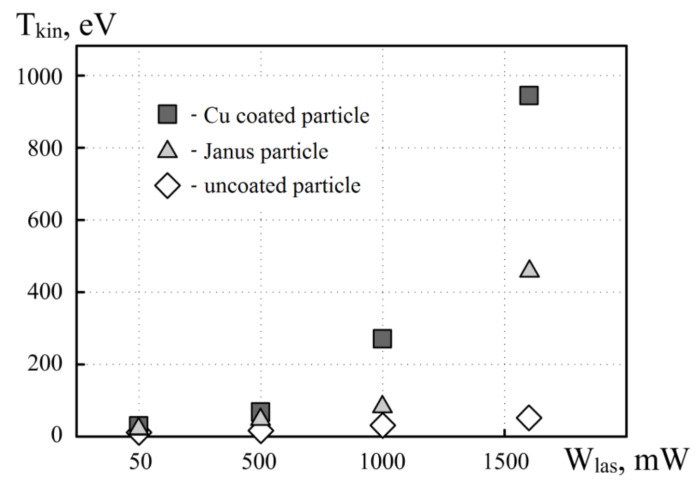
Kinetic energy of a single dust particle under the influence of laser radiation with a power of 50, 500, 1000 and 1600 mW for: uncoated particle, Cu-coated particle, and Janus particle.

**Figure 6 molecules-26-00561-f006:**
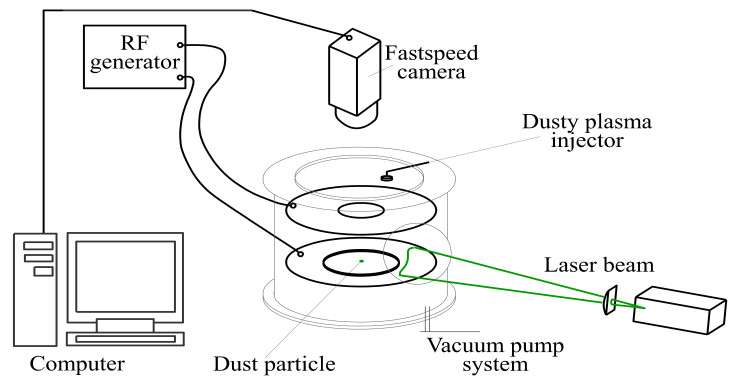
Scheme of the experimental setup.

## Data Availability

The data are available in supplementary material.

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
