# Peer review of "Dynamics of Active Brownian Particles in Plasma"

_molecules, 2021, doi:10.3390/molecules26030561_

Round 1

Reviewer 1 Report

The manuscript investigated the active Brownian motions of single particle in RF discharge plasma. Regrettably, however, the writing and organization of this paper are not good. More importantly, the manuscript only presented little experimental data, which does not contain enough scientific information that deserves publishing on the Molecules Journal. Hence, I do not recommend it for publication. The author should consider it carefully with the following suggestions:

  1. The manuscript didn’t present sufficient data to support the conclusion sentence in abstract: “The movement of the active particle is a superposition of chaotic motion and self-propulsion”. The passivity Brownian motion (chaotic motion) for the single particle isn’t measured and presented in this experiment.
  2. In introduction, the meaning and application of this investigation should be introduced.
  3. The cover area of iron in “Janus” particle should be tested and presented.
  4. The dynamic resolution of fastspeed camera, error analysis of experimental system and experimental frequency should be provided.
  5. In analysis and discussion, the authors only monotonously stated the results shown in these figures. A thorough discussion regarding the new phenomena they have found and the academic contributions should be provided.
  6. There are many typos and grammar mistakes in this manuscript. The English quality of the manuscript requires improvement. The authors are recommended to ask a native English speaker to proofread their manuscript.

Author Response

We thank the Reviewer for raising a number of important questions. We have addressed all the points and have highlighted the relevant corrections in the current version of the paper to comply with all comments and recommendations, given by Reviewer.

Below is our response to the issues raised in the review (printed in italics)

Response to Referee 1.

The manuscript investigated the active Brownian motions of single particle in RF discharge plasma. Regrettably, however, the writing and organization of this paper are not good. More importantly, the manuscript only presented little experimental data, which does not contain enough scientific information that deserves publishing on the Molecules Journal. Hence, I do not recommend it for publication. The author should consider it carefully with the following suggestions:

  1. The manuscript didn’t present sufficient data to support the conclusion sentence in abstract: “The movement of the active particle is a superposition of chaotic motion and self-propulsion”. The passivity Brownian motion (chaotic motion) for the single particle isn’t measured and presented in this experiment.

The following text has been added to the manuscript (page 3/8):

"The observed characteristics of motion at a high intensity of laser radiation are in good agreement with the theoretical model [12].The motion of homogeneous particles corresponds to "noise-free motion mode of a circle swimmer with constant self-propulsion in a constant spatial trap" (see Fig. 2 (a), (b) from [12]), while the motion of Janus particles partially corresponds to "noise-free motion mode of a circle swimmer with temporally varying self-propulsion force in a constant spatial trap" (Fig. 3 from [12]). Nevertheless, real trajectories of Janus particles differ from the ideal ones described by the model [12]. In both cases, the trajectories consist of arcs, however, the connection between arcs in the experiment is more chaotic than in the model [12]. "

  1. In introduction, the meaning and application of this investigation should be introduced.

The following text has been added to the manuscript (page 1/8):

"The novelty of the work is in comparing the motion of particles with different surface properties, which are placed in the same plasma medium."

  1. The cover area of iron in “Janus” particle should be tested and presented.

SEM photographs of the Janus particles are shown in Figure 1 (c), (e). The modified surface area was £50%  of the particle surface area (page 7/8).

            The real proportion of the coated part of the Janus particles was in the range of 30-50%. No separation was performed in this experiment. However, in our experiments, the nature of the movement of the Janus particles had characteristic features that are described in the manuscript.

  1. The dynamic resolution of fastspeed camera, error analysis of experimental system and experimental frequency should be provided.

            Characteristics of the camera: CCD-matrix is 1440´1440px, resolution is 14,6 μm/pixel, frequency is 400frames/sec. Corresponding corrections are added to the manuscript on page 6/8.

In our previous studies with the same experimental conditions [15], [18] we have shown that the systematic errors, that can arise during reconstruction of video, are within subpixel accuracy, so that errors are less then symbols in the graphs.  

  1. In analysis and discussion, the authors only monotonously stated the results shown in these figures. A thorough discussion regarding the new phenomena they have found and the academic contributions should be provided.

The novelty of this manuscript is in the comparison of the parameters of motion of three different types of particles in an identical electrostatic trap. The particles differed only in the surface parameters. Their size and weight were almost the same. The presented work is experimental. A detailed analysis of the mechanisms of motion is yet to be done in the future.

  1. There are many typos and grammar mistakes in this manuscript. The English quality of the manuscript requires improvement. The authors are recommended to ask a native English speaker to proofread their manuscript.

            We did our best to improve English in the manuscript, so we hope that now it fits journal standarts better. 

Reviewer 2 Report

In this manuscript, the authors study the dynamics of active particles in dusty plasmas. Three types of particles are investigated, including MF particles, MF particles with Cu coatings, and Janus particles coated partially. The trajectories are recorded using video cameras. Furthermore, the authors study the dependence of the MSD and linear displacement on the laser power. The preparation of Janus particles is described in details. The analysis of the dynamics of individual particles is rather straightforward. The discussion of the mechanism is, however, speculative. Nevertheless, as one of the first a few attempts to study the dynamics of active particles in dusty plasma, the manuscript is worth publishing for the reference for future work.

Author Response

We thank the reviewer for the work done. Several changes have been added to the updated version of the manuscript.

Reviewer 3 Report

I did not find many issues with this manuscript. So I recommend publication as it is more or less.

Its not clear how the mean squared displacement in fig. 3 was calculated - the authors should explicitly state the formulas used so that readers can make sense why the value reduces after initially increasing - since MSD is a non-negative quantity, particles doing random turns should not affect its magnitude, which should continue to increase. So some transparency there will make things clearer.

It is clear that uncoated, copper coated and Janus particles attain different charge levels in the plasma at different laser powers. Although the intention of the authors was to report the novelty of hte motion of Janus particles in plasmas, they should also try to estimate the charge on each type of particle to make sure it corroborates Fig. 5 (Kinetic temperature) rise of coated and Janus particles compared to uncoated.

Since its a study that reports the trajectories of different types of particles and a lot remains to be understood - what is the charge on the grains? how does that affect the motion of the Janus particles? do both hemispheres of the Janus particle attain the same charge? Can one predict the motion and average trajectories of Janus particles (or other particles) using simulation?

considering these open questions that this, potentially impactful study, opens up - the authors should consider making available their trajectory data to allow alternate theoretical interpretations of their experiments. Especially, the authors claim about self-propulsion seems weak to me, since there is no active control of the Janus particle - it merely obeys the laws of physics dictated by its electric charge and the response to the laser manipulation. 

I agree with the measurements that the authors report but I'd like to seem room for more theorizing in the future instead of conclusively saying this is the mechanism. Revising the language to include this point is suggested.

Overall, I recommend publication.

Author Response

We thank the Reviewer for raising a number of important questions. We have addressed all the points and have highlighted the relevant corrections in the current version of the paper to comply with all comments and recommendations, given by Reviewer.

Below is our response to the issues raised in the review (printed in italics)

Response to Referee 3.

  1. Its not clear how the mean squared displacement in fig. 3 was calculated - the authors should explicitly state the formulas used so that readers can make sense why the value reduces after initially increasing - since MSD is a non-negative quantity, particles doing random turns should not affect its magnitude, which should continue to increase. So some transparency there will make things clearer.

            The following text has been added to the manuscript (page 4/8):

            "To analyze the motion of particles, we calculated their mean square displacement (MSD), which is described by the formula <r2(t)>=<[r(t)-r(0)]2>, where vector r(0) is the initial position of the particle, and vector r(t) is the position of the particle at time t [16]. In the single-particle case, averaging is carried out over time. The <r(t)>2 graph is shown in Figure 3. "

As seen in Figure 3, MSD is a non-negative quantity. In the presence of a trap, the MSD parameter can have a periodic nature (see Fig. 4 (b) from [17] at a = 1).

  1. It is clear that uncoated, copper coated and Janus particles attain different charge levels in the plasma at different laser powers. Although the intention of the authors was to report the novelty of the motion of Janus particles in plasmas, they should also try to estimate the charge on each type of particle to make sure it corroborates Fig. 5 (Kinetic temperature) rise of coated and Janus particles compared to uncoated.

Since its a study that reports the trajectories of different types of particles and a lot remains to be understood - what is the charge on the grains? how does that affect the motion of the Janus particles? do both hemispheres of the Janus particle attain the same charge?

            The novelty of this manuscript is in the comparison of the parameters of motion of three different types of particles in an identical electrostatic trap. The particles differed only in the surface parameters. Their size and weight were almost the same. The presented work is experimental. A detailed analysis of the mechanisms of motion is yet to be done in the future. As for the charge, in accordance with common models, the value of the charge of dust particles in the rf discharge depends on the surface area [10], [11], and does not depend on the surface material. Therefore, the particle charge can be considered constant in a first approximation. In our opinion, photophoretic force plays the main role in this case, and its magnitude depends on the surface material.

Corresponding corrections added to the manuscript on page 2/8.

  1. Can one predict the motion and average trajectories of Janus particles (or other particles) using simulation?

  The following text has been added to the manuscript (page 3/8):

"The observed characteristics of motion at a high intensity of laser radiation are in good agreement with the theoretical model [12].The motion of homogeneous particles corresponds to "noise-free motion mode of a circle swimmer with constant self-propulsion in a constant spatial trap" (see Fig. 2 (a), (b) from [12]), while the motion of Janus particles partially corresponds to "noise-free motion mode of a circle swimmer with temporally varying self-propulsion force in a constant spatial trap" (Fig. 3 from [12]). Nevertheless, real trajectories of Janus particles differ from the ideal ones described by the model [12]. In both cases, the trajectories consist of arcs, however, the connection between arcs in the experiment is more chaotic than in the model [12]. "

  1. Considering these open questions that this, potentially impactful study, opens up - the authors should consider making available their trajectory data to allow alternate theoretical interpretations of their experiments. Especially, the authors claim about self-propulsion seems weak to me, since there is no active control of the Janus particle - it merely obeys the laws of physics dictated by its electric charge and the response to the laser manipulation.

            We have added the relevant videos as supplementary materials (see [14]).

  1. Especially, the authors claim about self-propulsion seems weak to me, since there is no active control of the Janus particle - it merely obeys the laws of physics dictated by its electric charge and the response to the laser manipulation.

I agree with the measurements that the authors report but I'd like to seem room for more theorizing in the future instead of conclusively saying this is the mechanism. Revising the language to include this point is suggested.

            Of course, the observed phenomena can be explained by different theoretical interpretations, for example mechanical models, parametric instability models and/or others. We used one of the modern concepts - the concept of active particles. A similar approach was used in works [9], [12], [13].

Round 2

Reviewer 1 Report

The author didn’t carefully modify the manuscript based on the reviewer’s comments. Therefore, this manuscript should be definitely rejected to keep the journal's reputation.

  1. The response to comment 1 is opposite to the conclusion sentence in the abstract: “The movement of the active particle is a superposition of chaotic motion and self-propulsion”. The “noise-free” in reference [12] means ignore the thermal noise or ignore the passivity Brownian motion. The Cu coated particle movement is in good agreement with the theoretical model (Fig.2 (a) from [12]). This phenomenon suggests that the movement of Cu coated active particle move directionally and periodically, not a superposition of chaotic motion and self-propulsion. Moreover, the movement of Janus particle is chaotic which opposite to the definition of active Brownian.
  2. The application of this investigation is unclear.
  3. The experiment is not rigorous. The movement of the Janus particle is caused by the iron coating and the particle coating together. The movement of iron fully coated particle should be supplemented.
  4. Whether the experimental system or the Janus particle is no fundamentally different from those in reference [6]. The conclusion that different surface properties result in different movement is obvious. Moreover, the authors didn’t explain and predict the movement of the Janus particle in principle. The real proportion of the coated part of Janus particles was not presented, which hinder the experimental reproduction and theoretical research in the future.
